# Transferring Reasoning Capabilities between LLMs operating via Curriculum Learning Policy

**Leonardo Ranaldi** *name.surname@ed.ac.uk*
*University of Edinburgh*
*University of Rome Tor Vergata*

**Giulia Pucci** *name.surname@abdn.ac.uk*
*University of Aberdeen*

**Fabio Massimo Zanzotto** *name.surname@uniroma2.it*
*University of Rome Tor Vergata*

**Reviewed on OpenReview:** *https://openreview.net/forum?id=zPKqyjmyEQ*

## Abstract

In-context *reasoning methods*, exemplified by Chain-of-Thought (CoT) *(et alia.,)* empower the reasoning abilities of large language models (LLMs), eliciting them to solve complex reasoning tasks step-by-step. Nevertheless, the capacities to deliver robust CoT explanations arise only in models with billions of parameters, representing a barrier to entry for many users forced to operate on a smaller model scale, i.e., Small Language Models (SLMs). Even though many companies are releasing LLMs of the same family with a reduced number of parameters, these models sometimes produce misleading answers and are unable to deliver accurate step-wise reasoned answers. This paper proposes a method to transfer step-wise reasoning over SLMs by operating via Instruction-tuning (IT) on synthetic demonstrations delivered in a pedagogically motivated manner. In particular, firstly, we propose aligning step-wise reasoning capabilities via IT using Demonstrations "taught" by LLMs teacher to SLMs students. Then, we operate via *Curriculum Learning*, a pedagogically motivated learning method that improves the IT phase. We analyse the impact on the downstream performances of four question-answering benchmarks. The results show that SMLs can be instructed to reason via Demonstrations delivered by LLMs. We move a step further in research: conceiving SLMs as human learners, we expose them to a CL teaching-based approach, obtaining better results on downstream performances.

## 1 Introduction

In-context *reasoning methods*, exemplified by Chain-of-Thought (CoT) *(et alia.,)* empower the reasoning abilities of large language models (LLMs), eliciting them to solve complex reasoning tasks step-by-step. These methods enable large language models (LLMs) to deliver multi-step, controlled reasoning Kojima et al. (2023); Wei et al. (2022), achieving outstanding results in commonsense, symbolic, and mathematical reasoning tasks. LLMs achieve all these results with at least several billions of parameters, such as GPTs OpenAI (2024), Llamas Touvron et al. (2023); Grattafiori et al. (2024) and Mistral MistralAI (2023).

In contrast, Small Language Models (SLMs) break down the problems and deliver step-wise answers less effectively. Although these models are highly functional across different tasks, the CoT mechanism consistently benefitted only models at a certain threshold scale or through costly and time-consuming post-training. SLMs are crucial in fostering research since these are smaller versions of LLMs that are often open-source and accessible to most researchers, e.g., Llama-3-1b and Llama-3-8b Grattafiori et al. (2024). However, these SLMs produce illogical answers when prompted under the CoT framework.

We propose an approach to align the reasoning abilities of the SMLs (students) with the LLMs (teachers) via Instruction-tuning-CoT (*IT-CoT*), that is, an instruction tuning over synthetic CoT Demonstrations delivered from larger models (Figure 1), following teacher-student pipeline Ranaldi & Freitas (2024a).

Concerning the foundation teacher-student approach Magister et al. (2023), we move a step further by introducing the *IT* via CoT, and, concerning Ranaldi & Freitas (2024a), we improve the strategies to expose the student to examples in a reasonable, pedagogically-motivated order Ranaldi et al. (2023), adapting Curriculum Learning Bengio et al. (2009). Starting from the idea that humans acquire first elemental concepts and then, gradually, more complex ones, Bengio et al. (2009) proposed Curriculum Learning (CL) and demonstrated its benefits in several tasks Ranaldi et al. (2024). We adopt this idea to reorder the *IT* Demonstrations in a meaningful way. Hence, we evaluate the reasoning chains that are answers delivered by teachers via CoT prompting to elicit student learning.

This leads to the target research questions, which are the focus of this paper: *RQ1 - How does IT-CoT via synthetic Demonstrations impact the reasoning abilities of students' models? And what is the effect of step-wise Demonstrations delivered with the CoT process? RQ2 - How important is reasoning chain valuation to facilitate the presentation of demonstrations during IT?*

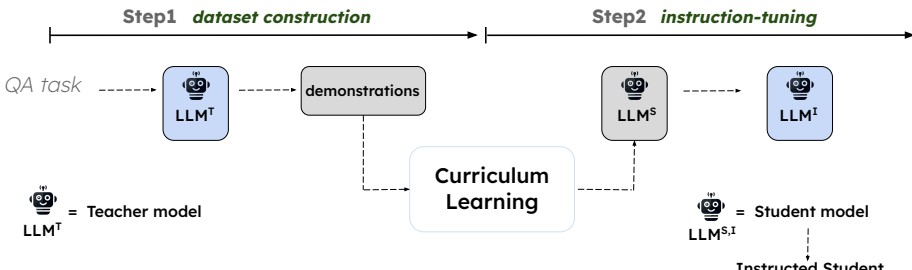

Figure 1: In Instruction-tuning, the smaller models instruct themselves using the synthetic demonstrations generated by the larger models. Hence, we elicit a larger model to deliver step-wise reasoned answer solutions. Moreover, we evaluate the reasoning chain using Curriculum Learning metrics to facilitate the instruction phase and expose the Demonstrations in a meaningful way.

We answer these questions by selecting smaller LLMs (Llama-3-1b, -8b as students and larger LLMs (Llama-3-70b, and GPT-4) as teachers. We perform a comprehensive analysis using four question-answering benchmarks. We use Llama-3-70 and GPT-4 to deliver Answers at the core of the Demonstrations (see Figure 1) to instruct Llama-3-1 and -13. Furthermore, we evaluate the complexity of the reasoning chains in generated answers to expose the students to Demonstrations delivered by teachers. Hence, we propose a metric based on informativeness comprehensibility used as a pivot in the *IT* phase.

Behind a wide analysis, we show that the *IT* and *IT-CoT* approaches on Demonstrations instruct students, and they outperform baseline SLMs in all proposed benchmarks. In addition, the students exposed to the Demonstrations via the CL approach outperformed students instructed via non-CL.

Our findings can be summarized as follows:

**i)** The *IT* of SLM students via Demonstrations delivered by an LLM teacher outperformed the baselines in terms of downstream performance. The SLMs instructed via Demonstrations consistently outperformed the baselines defined by non-tuned SLMs on the four proposed question-answering benchmarks.

**ii)** The CL-based *IT* approach outperforms standard *IT*. Llama-3-1 and Llama-3-8, instructed via the CL method, outperform the instructed models without CL.

**iii)** Finally, the CL method favours the alignment of CoT abilities within the family. Llama-3-1 and Llama-3-8 were exposed to CL Demonstrations produced by Llama-3-70, which outperformed students instructed by GPT-4 teachers in other SMLs as well.

## 2 Method

We propose three steps to align the reasoning abilities of smaller Language Models using further knowledge generated by larger Language Models, as shown in Figure 1. In the first part, there is an annotation phase where the large language models (LLMs) systematically prompt generate outputs (Section 2.1). The outputs will be the core of Demonstrations used during the Instruction-tuning *IT* phase from the smaller Language Models, presented in Section 2.2. However, the Curriculum Learning approach is behind the *IT* phase, where the Demonstrations are reorganized following our measure introduced in Section 2.3.

### 2.1 Teacher Model

As teacher model, we selected the largest Llama version Grattafiori et al. (2024), that is, Llama-3-70b, and in terms of comparison, GPT-4[1] OpenAI (2024). We selected GPT-4 because it generates high-quality data with and without the CoT prompting approach. Meanwhile, Llama-3-70b because it has smaller versions that can be used as students of the same family (presented in Section 2.2), and these smaller versions obtain remarkable results despite the reduced number of parameters.

Hence, we proposed the following prompt in a zero-shot scenario:

```
Choose the answer to the question only from
options A, B, C, D.
Question:  <Question>
Choices:
A) <Option1>
B) <Option2>
C) <Option3>
D) <Option4>
Answer:  Let's think step by step
```

Input prompts have a generic structure, but behind `"Answer:"` we insert the formula `"Let's think step by step"` that is shown in Table 7.

Following the annotation process performed by LLMs, the answers generated by teachers models that are the annotations have been used to construct the Demonstrations (see Table 1).

### 2.2 Student Model

Several SLMs have been fine-tuned for instruction-following Taori et al. (2023) and reinforcement learning with human feedback. However, whatever the techniques, the smaller Language Models do not seem able to break down a given task and deliver a step-wise solution.

Recent work proposes techniques of knowledge distillation and skill-refinement to teach SLMs to produce reasoned answers Ranaldi & Freitas (2024b). Following this idea, we analysed the impact of synthetic *IT* from Demonstrations provided by different teachers. Hence, we chose two SLMs students, Llama-3-1b and Llama-3-8b. In both cases, we used "instruct" versions, i.e., Llama-3-1-chat and Llama-3-8-instruct (versions detailed in Table 8), which we will refer to in the paper as Llama-3-1 and -8.

Student models are evaluated before and after the IT phases. In particular, we propose the standard approach where the students are instructed via Demonstrations delivered arbitrarily and in the Curriculum Learning manner (introduced in Section 2.3). The structure of IT arises from Alpaca Taori et al. (2023), which operates via Demonstrations consisting of an instruction that, in our case, is fixed, i.e., `Choose the answer to the question only from options A, B, C, D.`, an input which is the question, and an expected output which, in our case, are the output generated by the LLMs teachers. Table 1 shows an example of input. Additional details about the IT steps are provided in Section 3.2.1.

---

[1]We use *GPT-4-turbo*, however in the rest of work we will use only GPT-4

---

**Instruction**

Choose the answer to the question only from options A, B, C, D.

---

**Input**

*Question:*  Which form can be made into a triangle?
Choices:
A) ice
B) steam
C) water
D) salt water

---

**Output (that is the Answer produced by teacher)**

*Answer:*
A) ice:  Ice is a solid form of water.  In solid form, substances can be
molded or shaped into various forms, including triangles.  If you had a
block of ice, you could carve or mold it into a triangular shape.
B) steam:  Steam is the gaseous form of water.  Being a gas, it doesn't
have a fixed shape.  You can't shape steam into a triangle without
containing it within some form of solid boundary, and even then, the
steam itself isn't triangular – the container is.
C) water:  Water is a liquid that takes the shape of its container.  You
can pour water into a triangular container, and while the volume inside
would be triangular, the water itself isn't inherently triangular.

D) salt water:  This is essentially the same as option C, but with salt
dissolved in it.  Being a liquid, it takes the shape of its container.

Given these considerations, the only substance among the options that can
be shaped into a triangle by itself is **A) Ice**.

---

Table 1: Example of Demonstration. The structure is composed by: **Instruction**, **Input** and **Output**. In this case, the last part is the **Answer** produced by GPT-4 CoT prompted.

## 2.3   Curriculum Learning

Aligning the teacher-student reasoning abilities via synthetic Demonstrations delivered by step-wise answers provided by teachers, CoT prompted is a promising technique. However, some aspects need clarification: what constitutes an answer containing a good reasoning chain and how to evaluate it to optimize the IT phase. Following the Curriculum Learning (CL), where training algorithms can achieve better results when training data are presented according to the model's current skills, Bengio et al. (2009). We propose a method for evaluating the reasoning chain present in the CoT Answers (that represent the outputs of CoT Demonstration) using two fundamental properties: (1) comprehensibility, that is, the comprehensibility of a text according to metrics proposed by Talburt (1986), and (2) informativeness, that is, every step of the chain provides new information that is useful and informative for deriving the generated answer. We apply this metric to the CoT Answers provided by the teachers; then, we reorder the demonstrations according to our measure.

**Informativeness**   To quantify the effectiveness of each step contributing novel information beneficial for deriving the final Answer $A$, we propose an assessment based on the Entropy and Information Gain (IG). The Entropy, represented by $H(S)$, evaluates the unexpected within a given sequence $S$, where $S_i \in A$. The

entropy is given by:

$$H(S) = -\sum_{w \in S} p(w) \log_2 p(w) \tag{1}$$

where $p(w)$ denotes the probability of token $w$ occurring in the sequence. Hence, we compute the IG between a previous $S_{\text{prev}}$ and a current sequence $S_i$ as:

$$IG(S_{prev}, S_i) = H(S_{prev} + S_i) - H(S_{prev}) \tag{2}$$

This metric quantifies how much new information the current step adds relative to the cumulative content previously considered. To obtain a comprehensive measure, we calculate the average IG across the different steps as follows:

$$d_I(A_i) = \frac{1}{N} \sum_{i=1}^{N} IG(S_{prev}, S_i) \tag{3}$$

where $N$ represents the total number of steps in the Answer or the sequences $S_i$. We calculate this value for each answer $A_i$ and obtain the maximum $d_{I_{max}}$ and the minimum $d_{I_{min}}$ scores. Finally, we normalize these values:

$$\hat{d}_I(A_i) = \frac{d_I(A_i) - d_{I_{min}}}{d_{I_{max}} - d_{I_{min}}}, \forall i \in [0, |D|]. \tag{4}$$

where $|D|$ are all answers to a specific benchmark.

**Comprehensibility**   Typical factors for measuring comprehensibility are Speed of perception, Perceivability in peripheral vision, Reflex blink technique, Eye movements, Cognitively motivated features, and Word difficulty. However, it is not always possible to capture all these features.

Hence, we used the Flesch-Kincaid metric Talburt (1986). This metric is used to assess the comprehensibility of a text. It is based on the length of sentences and words within a text and provides a score that indicates the text's difficulty level. The lower the score, the easier it is to read and comprehend the text. The formula for calculating the Flesch-Kincaid Grade Level score is as follows:

$$d_C(A_i) = 0.39 \frac{Avg(d_L(A_i))}{100} + 11.8 \frac{Avg(d_L(w_i))}{100} - 15.59 \tag{5}$$

where $Avg(d_L(A_i))$ average answer length is the number of words in a sentence divided by the number of sentences, and $Avg(d_L(w_i)$ is the average word length, i.e. is the number of syllables per word divided by the number of words. The value 0.39 is used to scale the effect of the average sentence length to compare it to the effect of the average word length, weighted by 11.8. The final score is then adjusted by subtracting the value of 15.59, which adjusts the score scale to match the grading levels used in education more closely. We calculate this value for each Answer $A_i$ and obtain the maximum $d_{C_{max}}$ and the minimum $d_{C_{min}}$ scores. Finally, we normalize these values:

$$\hat{d}_C(A_i) = \frac{d_C(A_i) - d_{C_{min}}}{d_{C_{max}} - d_{C_{min}}}, \forall i \in [0, |D|]. \tag{6}$$

**Constructing the CL-Demonstration**   We gather the annotations (answers) delivered by the CoT-prompted teachers (as explained in Section 2.1), and we estimate the informativeness $\hat{d}_I(A_i)$ and complexity $\hat{d}_C(A_i)$ for each answer $A_i, \forall i \in |D|$.

Then we merge the two values in:

$$d_{IC}(A_i) = \hat{d}_I(A_i) + \hat{d}_C(A_i) \tag{7}$$

We use $d_{IC}(A_i)$ as a pivot value to reorder the Answers provided by the teachers. The Answers (which form the output of the Demonstrations) will be delivered in the Instruction-tuning phase to the students in ascending order with respect to the value $d_{IC}(A_i)$. These heuristics are very lightweight: using only 16GB of memory, we can process up to 20k Responses per second to produce the informativeness and comprehensibility metrics.

# 3 Experimental Setup

To make the experiments comparable with state-of-the-art models, we use four benchmarks (introduced in Section 3.1) that are generally used to assess the abilities of Large Language Models (LLMs). Moreover, to conduct the Instruction-tuning phase on the Small Language Models (SMLs), we use two approaches: the first one is presented in Section 3.2, which we call Instruction-tuning on Demonstrations; the second is based on the Curriculum Learning (CL) approach where the students are exposed to CL-Demonstrations that are Demonstrations reordered in a CL way, as exemplified in Section 2.3. All code is available in the supplementary material, to be released if accepted.

## 3.1 Data

**General Commonsense Reasoning**  We evaluate the models' ability to perform general reasoning on the CommonSenseQA Talmor et al. (2019) (CSQA) and OpenBookQA Mihaylov et al. (2018) (OBQA). CommonSenseQA is one of the best-known datasets of answers to multiple-choice questions dealing with different types of general commonsense knowledge. OpenBookQA is a resource that contains questions requiring multi-step reasoning, common knowledge, and rich text comprehension. It is inspired by high school-level open-book exams in physics and biology, aiming to assess human comprehension and application of foundational concepts.

**Physical Interaction Reasoning**  We evaluate the models' ability to perform physical reasoning on the Interaction Question Answering (PIQA) Bisk et al. (2019). It is a resource consisting of everyday situations with typical and atypical solutions.

**Social Interaction Reasoning**  We evaluate the models' ability to perform social reasoning on the Social Interaction Question Answering (SIQA) Sap et al. (2019). It is a benchmark focusing on reasoning about people's actions and social implications. The actions in Social IQa cover various social situations and candidates for plausible and not plausible answers.

**Splitting Details**  Since a test split for all benchmarks is not always available open-source, we adopt the following strategy: we use 4000 examples with equally distributed target classes as training data and the validation versions found on huggingface as test data. We performed this split to observe the impact of the responses provided by the teacher models on different benchmarks. The same is true for validation, since we needed open-source and reproducible data to conduct a detailed evaluation of the student models. In Table 9, we report the quantitative information, global, and splitting ratios. The data are fully accessible and open-source, as in Table 10.

## 3.2 Teaching to Reason

We selected Llama-3-70 and GPT-4 as the teachers (introduced in Section 2.1). Consequently, the LLMs are prompted in the zero-shot scenarios, as shown in Table 6 and Table 7.

We selected Llama-3-1 and Llama-3-8 Grattafiori et al. (2024) as student models (as described in Section 2.2). Therefore, the students models are Instruction-tuned via Demonstrations, as introduced in Section 3.2, and via CL-Demonstrations, as explained in Section 2.3. Table 1 shows a Demonstration containing the Instruction, Input, and, as Output, the Answer-delivering CoT, an output generated by GPT-4 CoT-prompted.

### 3.2.1 Models Setup

We conduct the Instruction-tuning phases using QLoRA proposed by Dettmers et al. (2023). This approach allows tuning to be conducted while reducing memory usage and preserving the performance. We follow the training approach proposed in Alpaca Taori et al. (2023) and trainoed the models for 3 epochs and set the learning rate to 0.00002 with 0.001 weight decay. We conducted our experiments on a workstation equipped with two Nvidia RTX A6000 with 48GB of VRAM.

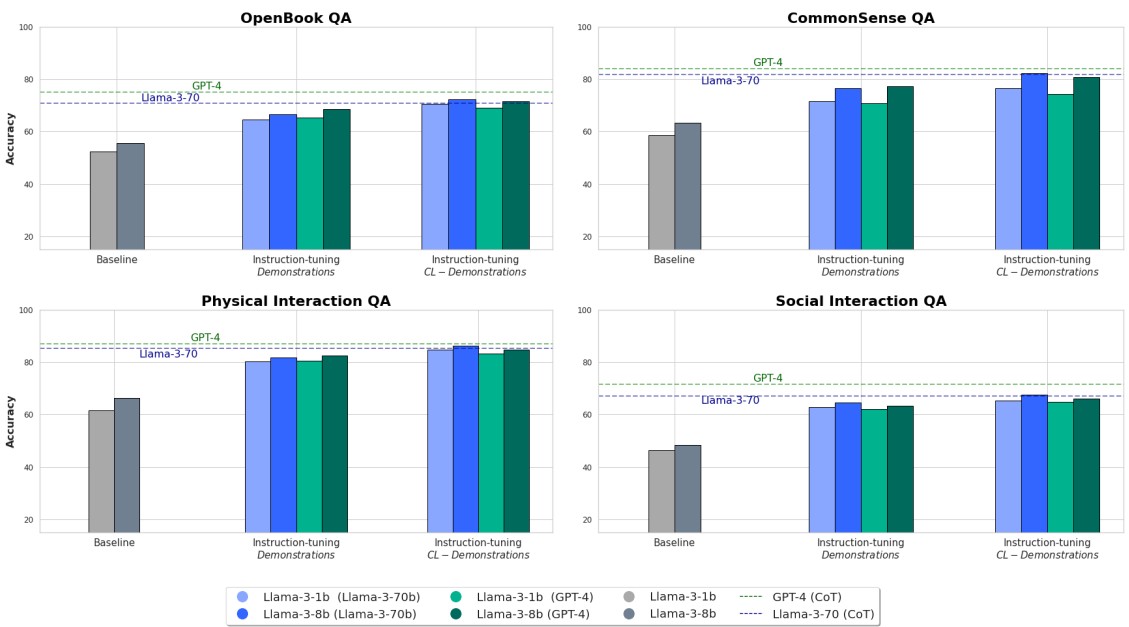

Figure 2: Accuracies (%) on benchmarks (Section 3.1) before Instruction-tuning (i.e., Baselines) and after on Demonstrations and CL-Demonstrations. Moreover, there are the teachers' performances also shown in Table 5

## 3.3 Evaluation

The most commonly used evaluation methods for question-answering tasks are language-model probing, in which the option with the highest probability is chosen Brown et al. (2020), and multiple-choice probing, in which the models are asked to answer. The evaluation is performed with a function taking the maximum value and, in the second case, with string matching. The second method is widely used in recent evaluations because it applies to models such as GPT-4 and GPT-4 OpenAI (2024) where probability values cannot be accessed. In our experiments, we chose the latter to have a comparable and scalable pipeline. Hence, we performed a string matching between the generated outputs and the targets.

## 4 Results

Language Models that do not get it can be elicited to do it through the knowledge of teacher models. These conclusions can be observed in Figure 2, where we reported the downstream performances without the Instruction-tuning phase (see the Baseline) and the Instruction-tuning on Demonstrations. As discussed in Section 4.1, Small Language Models (SLMs) CoT prompted obtained weak results. In contrast, models that are instructed via Chain-of-Thought (CoT) Demonstrations, i.e., Demonstrations produced by CoT-prompted Large Language Models (LLMs), outperform non-instructed models (Section 4.2). However, although Demonstrations produced better students, the complete alignments between students and teachers are realized with the *Curriculum Learning* procedure, as discussed in Section 4.3. In particular, the students instructed via the CL (Instruction-tuning CL-Demonstration in Figure 2) outperformed the students instructed via standard Instruction-tuning.

Finally, the CL approach delivers the teacher-student family-alignment. In Figure 2 (horizontal lines), it is possible to observe the phenomenon of family-alignment between Llama-3-70 and Llama-3-1 and -13 in more detail in Section 4.4.

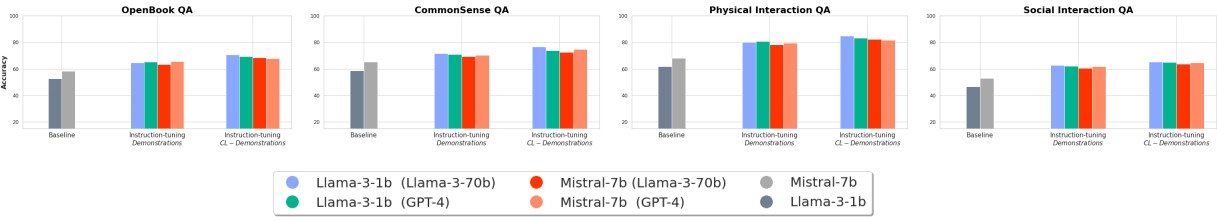

Figure 3: Accuracies of Llama-3-1 and Mistral-7 Instruction-tuned using setup proposed Section 3.

## 4.1 CoT-abilities of Small Language Models

Chain-of-Thought (CoT) prompts do not consistently deliver downstream performance improvements. SLMs, i.e., with fewer parameters, have not benefitted the prompting with the CoT mechanism. In particular, we evaluated performance on four question-answering benchmarks, described in Section 3.1, using two versions of Llama. Using a baseline demonstration produced via a zero-shot prompt (which we call "Baseline") and a CoT prompt (Table 6 and Table 7), we obtained the performances in Table 2.

The results confirm what Wei et al. (2022) have claimed about the limitations of the emergent CoT prompting abilities that are not observable in SLMs. Moreover, using CoT prompting leads to model confusion, resulting in the subsequent degradation of downstream results. It is possible to observe these phenomena in OpenBookQA (OBQA) and CommonSenseQA (CSQA) (down arrows in Table 2). In particular, there is a marked deterioration in Llama-3-1 (see ⇓), which has half the parameters of Llama-3-8 (see ↓).

The same behaviour was not observed for Physical- and Social-Interaction Question Answering (PIQA) and (SIQA). Not considering the nature of benchmarks, unlike the others, they are always question-answering multiple-choice-questions but have fewer possible choices, as shown in Table 9. In this regard, we hypothesise that the most controllable scenarios, where chain reasoning is limited to fewer options, are reasonable by SLMs elicited with CoT prompts.

| Benchmarks | Llama-3-1 | | Llama-3-8 | |
|---|---|---|---|---|
| | Baseline | CoT | Baseline | CoT |
| OBQA | **53.6** | 51.0⇓ | **58.7** | 57.0↓ |
| CSQA | **58.9** | 51.2⇓ | **63.8** | 61.2↓ |
| SIQA | 46.8 | *45.9* | 48.7 | 47.8 |
| PIQA | 62.0 | *63.9* | 67.2 | *69.0* |

Table 2: Accuracies of Llama-3-1 and Llama-3-8, both without further tuning, on testing data with the standard prompt (Baseline) (see Table 6) and CoT prompt (CoT) (see Table 7).

## 4.2 The Instruction-tuning Method

Instruction-tuning led by Large Language Models (teachers models), able to reason elicit the Smaller Language Models (students models) to do the same. This is shown in Figure 2. The student models after Instruction-tuning on synthetic Demonstrations delivered by teacher models outperformed the baselines in the four proposed benchmarks. While performances are conspicuous improvements overall, they have sensible variations. The teacher models have different characteristics. They consequently achieve different performances in the proposed benchmarks. Table 5 shows the performances in the zero-shot scenario (CoT prompting and not) on the data used to conduct the Instruction-tuning phase and on the same test set used to evaluate the proposed models.

Although the performances on the "training set" are different (performances of GPT-4 and the same for Llama-3-70 in Table 5), this bias does not affect the students. The Llama-3-1 and -8 with GPT-4 as teacher outperform the Llama-3-1 and -8 with Llama-3-70 as teacher only on OBQA. As far as CSQA and PIQA

are concerned, there is a balance that is not present in SIQA, where the students of Llama-3-70 outperform the others.

However, in the Instruction-tuning method, instruction is conducted using Demonstrations (composed of Answers provided by teachers) delivered arbitrarily. Therefore, we propose to study both the intrinsic complexity of the answers and their impact on the students' exposure. To this end, we introduce a CL-based instruction approach where demonstrations are delivered to students in a meaningful order (Section 4.3).

### 4.3 The Impact of Curriculum Learning

Instruction-tuning via Curriculum Learning Demonstrations elicits the reasoning abilities of students. The students gradually exposed to increasingly meaningful Demonstrations (CL-Demonstrations) learn better than those exposed to arbitrary Demonstrations. This is shown in Figure 2 (bars Instruction-tuned CL-Demonstrations), where Llama-3-1 and -13 consistently outperformed the other models.

The benchmarks where the most significant effects can be observed are CSQA and OBQA, with an increase in average accuracy scores of 6 and 5 points, respectively (also in additional evaluations in Appendix G). The same effects are less evident in PIQA and SIQA. One possible reason for this phenomenon might be tied again to the nature of the benchmarks, as hypothesised in Section 4.1. To analyse this phenomenon, we studied the components of the complexity measure proposed in Section 4.5.

### 4.4 The role of CL in family-alignment

Instruction-tuning via CL-Demonstrations still aligns students' reasoning abilities with family teachers, even as instruction decreases. In fact, from Figure 2, we can observe that the performances of students instructed via CL-Demonstrations delivered by teachers from the same family outperform the others.

Moreover, to validate our hypothesis of family alignment, we introduced Mistral-7b MistralAI (2023), a new SLMs with 7 billion parameters that outperforms the Llama-3-8 version on several benchmarks, as shown by MistralAI (2023). Specifically, we reproduced the experiments introduced in Section 4.2. In Figure 3, it can be seen that Llama-3-1 instructed on different types of Demonstrations delivered by Llama-3-70 almost consistently outperforms Mistral-7b. In order to confirm this speculation, we scaled the experiments on additional models chosen by family and number of parameters, presenting the results in Appendix H. The results discussed in this section, as well as those presented in the appendices, confirm that Demonstrations derived from in-family teachers have a more significant impact on student models than those from other sources.

### 4.5 Ablation Study

**The role of measures** The informativeness and complexity exposed to students in a meaningful order instructs better students. We conducted an Ablation study to estimate the impact of our evaluation measures proposed in Section 2.3. Hence, we reproduced the same configurations proposed in Section 4.2, but removed one of the components (informativeness and complexity presented in Section 2.3). The results in Table 4 show that students instructed on CL-Demonstrations ordered by comprehensibility and informativeness consistently outperform students instructed via Demonstrations. The results show that students trained on the Demonstrations sorted by informativeness are more productive in QA tasks with more choices. In comparison, complexity proved helpful in cases where the number of options is minor. Phenomenon manifested in CSQA and OBQA with 5 and 4 choices, and PIQA and SIQA with 2 and 3 choices.

**The Impact of Demonstration Quality** The quality of the demonstrations plays a crucial role. By filtering them using a heuristic based on string matching and GPT-4o annotations (as described in Appendix C), we analysed the effects of different types of training examples. As displayed in Table 3, models instructed with demonstrations ordered via our CL metrics consistently outperform others, even when the demonstrations are incorrect or misleading. This suggests that the metric introduced in Section 2.3 has a beneficial effect regardless of the correctness of individual training examples.

| Method | OBQA | CSQA | SIQA | PIQA |
|---|---|---|---|---|
| *Instruction-tuning CL* | 71.2 | **80.2** | 84.3 | 66.4 |
| *Instruction-tuning CL (gold)* | **72.7** | **80.2** | **85.0** | **66.8** |
| *Instruction-tuning (gold)* | 70.3 | 78.5 | 82.6 | 65.4 |
| *Instruction-tuning CL (misleading)* | 60.0 ↑ | 63.0 | 48.9 ↑ | 68.4 ↑ |
| *Instruction-tuning (misleading)* | 59.2 | 62.6 ↓ | 46.5 ↓ | 67.0 ↓ |
| *baseline* | 58.3 | 63.1 | 47.9 | 67.3 |

Table 3: Accuracies of Llama-3-8, instructed using Instruction-tuning (NB without *CL*) and Instruction-tuning CL demonstrations as in Section 3, using misleading (incorrect) and gold (correct) demonstrations.

## 5 Related & Future Work

### 5.1 Learning from Natural Language Explanation

Current methods for conditioning models on task instructions and provided explanations for individual data points replace the ancient intermediate structures Hase & Bansal (2022) that used rationales Zhang et al. (2016) or inputs Narang et al. (2020); Talmor et al. (2020) to learn the models. Reasoning via the CoT builds upon prior efforts wherein explanations are viewed as intermediary constructs produced during inference Rajani et al. (2019). Our research stems from the studies of Ranaldi & Freitas (2024a;b). Particularly, we adopt the idea of an LLM teacher and a second, smaller LLM that takes a student's position Magister et al. (2023). Learning uses teacher-generated explanations, demonstrating prompt CoT downstream Li et al. (2023); Ho et al. (2023). Li et al. (2023) claims that massive demonstrations significantly improve performance over the single-sample approach Shridhar et al. (2023).

### 5.2 Large Language Models as a Teacher

Several papers have been published simultaneously, including those by Ranaldi & Freitas (2024a;b); Paul et al. (2024), and Saha et al. (2023) that prove the effect of transferring ability to produce CoT reasoning from larger to smaller models. Table 11 resumes all main points of these contributions.

Our work goes beyond the following ways: *i)* We propose a method for aligning CoT abilities via Instruction-tuning through Demonstrations produced by answers generated by larger models. *ii)* We study how to provide Demonstrations to students by proposing a measure for evaluating the answers provided and we analyse the alignment performance between in- and out-family models. *iii)* We propose an approach to improve the teachers and students' alignment by employing our evaluations to expose the students meaningfully.

### 5.3 Future Work

We plan to continue investigating ways to transfer models' capabilities by introducing parallel work heuristics to improve the structure of demonstrations by maximising abstraction abilities Ranaldi et al. (2025b), tuning models beyond single language Ranaldi & Pucci (2025) and modalities Ranaldi et al. (2025a).

## 6 Conclusion

We propose a method to enable step-wise reasoning over SLMs by introducing two mechanisms. First, we propose aligning CoT abilities via Instruction-tuning with the support of CoT Demonstrations delivered by LLMs teacher. Second, we use the Curriculum Learning to empower the tuning phase. We analyse the impact on the downstream abilities of four benchmarks. Results show that SLMs can be instructed to deliver robust, reasoned answers via Demonstration produced by LLMs. We move a step further in research: conceiving SLMs as human learners, we expose them to a CL teaching-based approach, obtaining better results on downstream performances.

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

## A Ablation study on CL-based Instruction-tuning

| Students | Benchmarks | | | |
|---|---|---|---|---|
| | **OBQA** | **CSQA** | **PIQA** | **SIQA** |
| *Llama-3-1 (Llama-3-70)* | | | | |
| *Arbitrary Teaching* | 64.7 | 71.6 | 80.2 | 62.8 |
| *Teaching via IC* | **70.5** | 76.5 | **84.8** | 65.3 |
| *Teaching via I* | 70.2⇑ | **77.2**⇑ | 81.2 | 61.8⇓ |
| *Teaching via C* | 66.4 | 69.7⇓ | 84.3⇑ | **66.2** |
| *Llama-3-1 (GPT-4)* | | | | |
| *Arbitrary Teaching* | 65.3 | 70.8 | 80.5 | 62.2 |
| *Teaching via IC* | **69.2** | **74.2** | 83.3 | **64.8** |
| *Teaching via I* | 68.5⇓ | 73.7⇑ | 79.6⇓ | 63.8 |
| *Teaching via C* | 66.3 | 69.8 | **83.9**⇑ | 65.7⇑ |
| *Llama-3-8 (Llama-3-70)* | | | | |
| *Arbitrary Teaching* | 66.5 | 76.5 | 81.9 | 64.5 |
| *Teaching via IC* | 72.3 | **82.2** | **86.2** | 67.7 |
| *Teaching via I* | **73.4**⇑ | 81.9⇑ | 80.7⇓ | 63.8 |
| *Teaching via C* | 67.9 | 76.6 | 84.3⇑ | **70.3** |
| *Llama-3-1 (GPT-4)* | | | | |
| *Arbitrary Teaching* | 68.5 | 77.3 | 82.6 | 63.3 |
| *Teaching via IC* | **71.6** | 80.5 | **84.9** | **66.1** |
| *Teaching via I* | 70.8⇑ | **81.7** | 81.9 | 62.7 |
| *Teaching via C* | 68.2⇓ | 78.5 | 82.3 | 65.9⇑ |

Table 4: Ablation study on our Instruction-tuning CL-Demonstrations approach.

| Benchmarks | Llama-3-70 | | GPT-4-o | |
|---|---|---|---|---|
| | **Baseline** | **CoT** | **Baseline** | **CoT** |
| **Training** | | | | |
| OBQA | 65.8 | 71.3 | 67.1 | **78.8** |
| CSQA | 75.0 | 79.6 | 80.4 | **85.6** |
| SIQA | 66.2 | 67.5 | 69.2 | **72.6** |
| PIQA | 82.9 | **86.0** | 83.5 | 85.8 |
| **Testing** | | | | |
| OBQA | 66.0 | 72.0 | 68.3 | **75.6** |
| CSQA | 73.4 | 81.8 | 81.4 | **84.0** |
| SIQA | 65.2 | 66.9 | 67.3 | **71.8** |
| PIQA | 83.8 | 85.6 | 85.2 | **86.5** |

Table 5: Accuracy (%) of Llama-3-70 and GPT-4 (teachers) on training and testing data with CoT prompt (CoT) and with the standard prompt (Baseline).

## B  Instruction used

---
**Zero-Shot**

```
Choose the answer to the question only from options A, B, C, D.
```
Question: Which animal gives birth to live young?
A) Shark
B) Turtle
C) Giraffe
D) Spider
```
Answer:
```
---

Table 6: Example of Zero-Shot prompting.

---
**Zero-Shot Chain-of-Thought**

```
Choose the answer to the question only from options A, B, C, D.
```
Question: Which animal gives birth to live young?

A) Shark
B) Turtle
C) Giraffe
D) Spider
```
Answer: Let's think step by step
```
---

Table 7: Example of Zero-Shot Chain-of-Thought prompting.

## C  Evaluation Metrics

We used a double-check to assess the accuracy of the responses delivered in the different experiments. In the first step, we used an exact-match heuristic (this was used for most of the evaluations, especially in cases of multiple-choice QA). However, since some experiments required a more accurate response check, we used GPT-4o as a judge.

---
**GPT-4o Evaluation Prompt**

Given the following "#Senteces", you are a decider that decides whether the "Generated Answer" is the same as the "Target Answer". If the output doesn't align with the correct answer, respond with '0', whereas if it's correct, then respond with '1'. *Please, do not provide any other answer beyond '0' or '1'.*
**#Senteces:**
Generated Answer: {model_result}
Target Answer: {correct_answer}.
---

## D  Models

| Model | Version |
|---|---|
| Llama-3-1 | meta-llama/Llama-3.2-1B-Instruct |
| Llama-3-8 | meta-llama/Meta-Llama-3-8B-Instruct |
| Llama-3-70 | meta-llama/Meta-Llama-3-70B |
| Mistral-7 | mistralai/Mistral-7B-Instruct-v0.2 |
| GPT-4o | OpenAI API (gpt-4o-2024-08-06) |

Table 8: In this table, we list the versions of the models proposed in this work, which can be found on huggingface.co. We used all the default configurations proposed in the repositories for each model.

## E   Training and Testing Data

|  | OBQA | CSQA | PIQA | SIQA |
|---|---|---|---|---|
| classes | 4 | 5 | 2 | 3 |
| **Training** # examples for each class | 1000 | 800 | 2000 | 1330 |
| **Test** # examples for each class | 125* ($\pm$ 8) | 235* ($\pm$ 11) | 924* ($\pm$ 18) | 640* ($\pm$ 19) |

Table 9: Characteristics Training and Test set of benchmarks proposed in Section 3.1. The * indicates that the number of examples are not perfect balanced, but the difference from the average is marginal.

| Name | Repository |
|---|---|
| CSQA Talmor et al. (2019) | `huggingface.co/datasets/commonsense_qa` |
| OBQA Mihaylov et al. (2018) | `huggingface.co/datasets/openbookqa` |
| PIQA Bisk et al. (2019) | `huggingface.co/datasets/piqa` |
| SIQA Sap et al. (2019) | `huggingface.co/datasets/social_i_qa` |

Table 10: In this table, we list the versions of the benchmark proposed in this work, which can be found on huggingface.co.

## F   Comparison with related works

| Work | Method | Teachers | Students |
|---|---|---|---|
| Magister et al. (2023) | SFT | PaLM GPT-4 | T5-small, -medium T5-large, -xxl |
| Li et al. (2023) | SFT | GPT-3 175B | OPT-1.3b |
| Shridhar et al. (2023) | SFT | GPT-3 175B | GPT-2 |
| Ho et al. (2023) | SFT | InstructGPT (text-davinci-002) | GPT-3 (ada,babbage,curie) |
| Ranaldi & Freitas (2024a) | IT | GPT-3.5 Llama-2-70 | Llama-2-7, Llama-2-13 |
| Ours | **IT** | **Llama-3-70b GPT-4** | **Llama-3-1b, -13b Mistral-7b** |

Table 11: Summary of methods, teacher and student models of previous work, we indicate Supervised Fine-tuning as (SFT) employed in most previous work.

## G  Evaluation on additional benchmark

| Method | Benchmarks | | | |
| --- | --- | --- | --- | --- |
| | GSM8K | MATH | AIME24 | MMLU-Redux |
| *Instruction-tuning CL* | 78.6 | **38.5** | 26.0 | **72.0** |
| *Instruction-tuning CL (gold)* | **80.2** | 37.0 | 22.8 | 70.7 |
| *Instruction-tuning (gold)* | 75.9 | 35.8 | 18.4 | 67.2 |
| *Instruction-tuning CL (misleading)* | 70.4 | 25.0 | 12.6 | 59.0 |
| *Instruction-tuning (misleading)* | 67.6 | 24.8 | 10.0 | 57.3 |
| *Baseline* | 73.8 | 30.0 | 16.0 | 60.2 |

Table 12: Results of different instruction-tuning strategies on GSM8K, MATH, AIME24, and MMLU-Redux using Llama-3-8b as base model.

## H  Evaluation on different Teacher-Student Settings

| Method | Benchmarks | | | | Additional QA Benchmarks | | | |
| --- | --- | --- | --- | --- | --- | --- | --- | --- |
| | GSM8K | MATH | AIME24 | MMLU-Redux | OBQA | CSQA | PIQA | SIQA |
| *IT-CL (GPT-4o)* | 78.6 | 38.5 | 26.0 | 72.0 | 71.2 | 80.2 | 66.4 | **84.3** |
| *IT-CL (Qwen2-72b)* | 79.2 | **39.0** | 25.0 | 73.1 | **72.8** | **82.6** | **68.5** | 83.2 |
| *IT-CL (Mixtral8x7b)* | 77.0 | 36.4 | 24.2 | 72.0 | 71.2 | 81.0 | 66.0 | 80.4 |
| *Baseline* | 73.0 | 28.0 | 10.0 | 58.6 | 58.0 | 62.9 | 66.5 | 48.2 |

Table 13: Instruction-tuning CL (**IT-CL**) with different teachers (GPT-4o, Qwen2-72b, Mixtral8x7b) compared on GSM8K, MATH, AIME24, MMLU-Redux, and QA benchmarks.

| Method | Benchmarks | | | | Additional QA Benchmarks | | | |
| --- | --- | --- | --- | --- | --- | --- | --- | --- |
| | GSM8K | MATH | AIME24 | MMLU-Redux | OBQA | CSQA | PIQA | SIQA |
| *IT-CL (GPT-4o)* | 76.9 | 38.0 | 22.8 | 70.2 | 71.4 | 81.0 | 66.1 | 83.6 |
| *IT-CL (Qwen2-72b)* | 78.1 | 37.2 | 21.8 | 70.4 | 71.0 | 80.5 | 67.5 | 82.1 |
| *IT-CL (Mixtral8x7b)* | **79.6** | **39.8** | **25.0** | **72.8** | **73.1** | **82.4** | **69.0** | **84.6** |
| *Baseline* | 72.8 | 28.2 | 10.0 | 58.3 | 59.2 | 63.5 | 67.2 | 48.6 |

Table 14: Instruction-tuning CL (**IT-CL**) with different teachers (GPT-4o, Qwen2-72b, Mixtral8x7b) compared on GSM8K, MATH, AIME24, MMLU-Redux, and QA benchmarks (second experimental setting).

