# OpenReview forum: "Transferring Reasoning Capabilities between LLMs operating via Curriculum Learning Policy"
_TMLR — Accepted by TMLR_

### Review · Reviewer_8eHQ · 2025-03-28

**Summary Of Contributions:**

This paper addresses the challenge of enabling SLMs to perform complex reasoning tasks.

The authors propose a method to transfer step-wise reasoning capabilities to SLMs through IT using synthetic demonstrations generated by LLMs.  They further enhance this process using Curriculum Learning to order the training demonstrations in a meaningful way.  The authors evaluate their approach on four question-answering benchmarks.

Their results indicate that SLMs can be effectively instructed to reason using demonstrations provided by LLMs.  Furthermore, the Curriculum Learning-based Instruction-tuning outperforms standard Instruction-tuning.  The study also finds that CL facilitates the alignment of Chain-of-Thought (CoT) abilities within the same model family.

**Audience:**

Yes

**Claims And Evidence:**

Yes

**Requested Changes:**

I would request the authors to :
1. Fix the typographical errors in the paper.
2. Add some results for Mathematics specific datasets

**Strengths And Weaknesses:**

**Strengths:**
1. The paper addresses a relevant problem in the field of Large Language Models, which is improving the reasoning capabilities of Small Language Models.
2. The proposed approach of using Curriculum Learning to enhance Instruction-tuning is novel and shows promising results.
3. The paper provides a comprehensive analysis using four different question-answering benchmarks.
4. The ablation study effectively demonstrates the impact of the proposed evaluation measures.

**Weaknesses:**
1. The paper has minor typographical errors throughout such as "Llama-3-1 and -13" in the introduction, Missing brackets in "Avg(dL(wi)" .
2. The paper does not include an evaluation of mathematics-specific datasets, which is a critical aspect of assessing model reasoning and CoT capabilities capabilities.

---

> ### Author Response · Authors · 2025-03-31
> **Response**
>
> Dear Reviewer 8eHQ,
>
> Thank you for your review. We really appreciate the fact that you liked the concept of teacher-student and conjunction with Curriculum Learning heuristics.
>
> Thank you very much for pointing out the typos.
>
> Regarding the testing of mathematical problems, we have integrated GSM8K, MATH and MMLU-Redux into the evaluation. While the former is one of the best-known in the domain of evaluating the mathematical capabilities of LLMs, the other two are new resources that are used very often to evaluate the latest LLMs released.  MATH is clearly related to mathematical topics, while MMLU-Redux is a corrected and refined version of MMLU and deals with multidisciplinary topics, including mathematics.
>
> | Method                                       | GSM8K  | MATH   | AIME24 | MMLU-Redux  |
> |---------------------------------------------|--------|--------|--------|-------------|
> | *Instruction-tuning CL*                     | 78.6   | **38.5** | 26.0  | **72.0**    |
> | *Instruction-tuning CL (gold)*              | **80.2** | 37.0   | 22.8  | 70.7        |
> | *Instruction-tuning (gold)*                 | 75.9   | 35.8   | 18.4  | 67.2        |
> | *Instruction-tuning CL (misleading)*        | 70.4   | 25.0   | 12.6  | 59.0        |
> | *Instruction-tuning (misleading)*           | 67.6   | 24.8   | 10.0 | 57.3        |
> | *baseline*                                  | 73.8   | 30.0   | 16.0  | 60.2        |
>
>
> These experiments were produced using the proposed method (we used Llama-3-8b as a base model and trained it on the various tuning strategies).

---

### Review · Reviewer_hrCx · 2025-04-27

**Summary Of Contributions:**

This paper proposes a method to enhance the reasoning capabilities of small language models by incorporating curriculum learning into the teacher-student distillation approach. The method has been experimentally validated on four multiple-choice question-answering benchmarks and the Llama series of models.

**Audience:**

Yes

**Broader Impact Concerns:**

Not involved.

**Claims And Evidence:**

No

**Requested Changes:**

Please see the major concerns in the Weaknesses.

Minor Concerns:
1. There is an issue with the subscript i in Formula 3.
2. The paragraph containing the description of Comprehensibility seems unrelated to this paper.

**Strengths And Weaknesses:**

**Strengths**
1. Introducing curriculum learning during the distillation process is quite intuitive.
2. The authors also provide comparisons of distillation performance across different teacher models, along with some analysis, such as the observation that models from the same family are more suitable for distillation.

**Weaknesses**
1. One major concern is that the authors' description of the motivation for solving the problem is incomplete. The reasoning capabilities of small models are not weak, especially in this era, where we see that R1-distill-Qwen-7B models significantly outperform the teacher models proposed in the paper (e.g., Llama3-70B, GPT-4) on the majority of reasoning datasets. Additionally, the authors mention that models smaller than 60B parameters benefit less from Chain-of-Thought (CoT) [1], but this is an outdated view from over two years ago. Due to the insufficient and outdated motivation, the study's use of Llama3-70B / GPT-4 models to distill Llama3-7B seems to lack significant purpose.
2. The second major concern is that the method is overly simple and lacks innovation. Teacher-student distillation and curriculum learning are well-established concepts that have been extensively studied in many works. The only difference in this paper is that the authors propose two metrics for measuring curriculum learning: Informativeness and Comprehensibility. However, these two metrics are also not entirely reasonable. For instance, the formula for Comprehensibility is overly simplistic, and a potentially better evaluation method might be to have a LLM assign scores. As for Informativeness, while it may measure changes in information gain, this does not necessarily correlate with the quality of CoT, such as whether it directly determines if the answer is correct.
3. The third major concern is that the experimental validation is not comprehensive. First, the dataset is limited to multiple-choice question-answering, which covers a very narrow scope. Second, the most commonly used reasoning tasks, such as mathematics and coding datasets, are not addressed in this paper. Furthermore, at the model level, the paper does not include mainstream model families such as the Qwen series. Finally, regarding the selection of teacher models, it is clear that neither GPT-4 nor Llama3-70B represents the strongest available teacher models. The insufficient experimental validation undermines the credibility of the conclusions drawn in this paper.

[1] Chain-of-thought prompting elicits reasoning in large language models

---

### Review · Reviewer_H517 · 2025-05-02

**Summary Of Contributions:**

This paper presents the idea of instruction-tuning a smaller language model with synthetic CoT demonstrations generated by a larger language model. The authors propose to perform curriculum learning via reordering demonstrations based on their designed metrics. Specifically, their metric quantifies the informativeness and comprehensibility of each demonstration. They evaluate their approach on 4 QA datasets, and perform training on Llama and Mistral models. The teacher models in their evaluation include models from the Llama family and GPT-4. Their evaluation shows that curriculum learning brings some performance gain, and picking a teacher model from the same family as the student model achieves better performance.

**Audience:**

Yes

**Broader Impact Concerns:**

No concern.

**Claims And Evidence:**

No

**Requested Changes:**

See the weakness section.

**Strengths And Weaknesses:**

Strengths:
1. The curriculum learning design of this work is interesting.

2. The discussion of using a teacher model from the same or different family is useful.

Weaknesses:

1. The novelty of this work is limited, and the current draft does not properly highlight the new design. The idea of using a larger language model to generate synthetic CoT demonstrations for training a smaller language model is not new, and has been studied in a bunch of prior works. Even if this work performs experiments using different models from prior works, the experimental results are expected and do not bring new insights. The paper should have instead focused more on curriculum learning and model family alignment between the student and the teacher model.

2. To validate the model family alignment hypothesis, the authors should evaluate on more model families other than Llama. For example, the authors can take different models from Gemma, Qwen or Mistral families, and see if the same conclusion holds there.

3. For curriculum learning, the authors should compare their approach to some simpler heuristics, e.g., reordering demonstrations according to their reasoning token counts. Does the proposed metric improve the performance over these baselines?

4. The authors should add experiments on standard reasoning benchmarks, such as GSM8K, MATH or harder tasks like AIME.

5. The writing of this paper should be improved. For example, the second paragraph of Section 4.5 include problematic pointers to sections.

---

> ### Author Response · Authors · 2025-05-03
>
> Dear Reviewer H517,
>
> We appreciate that you enjoyed the paper and that you observed the teacher-student analysis.
>
> In the next few lines, we address your remarks:
>
> 1. **curriculum learning and model family alignment between the student and the teacher model contribution**
>
> We definitely agree with you that the real contribution of the paper lies in the proposal of CL-based heuristics to build a solid teacher-student framework.
>
> Precisely for this reason, we clearly outlined in the introduction that we complement the previous ones and *‘..we improve the strategies to expose the student to examples in a reasonable, pedagogically-motivated order using CL..’* (reported from the paper). Same in the exemplification in the findings at the end of the introduction, where we emphasised the CL role and in-family dynamics.
>
> However, we will exemplify this better because,e as you also noted, the heart of the work is the CL approach.
>
>
> 2. Thank you for your interest and for giving us the opportunity to show this on other models. As we responded to Reviewer hrCx (we paste in the results), we experimented with different dynamics in further models and observed the following results in in-family and out-family dynamics. We also iterated on Gemma (as you advised).
>
> Please let us know if you have any further questions, and we will be happy to discuss them.
>
> ----
> | Method                                       | GSM8K  | MATH   | AIME24 | MMLU-Redux  | OBQA  | CSQA  | PIQA  | SIQA  |
> |---------------------------------------------|--------|--------|--------|-------------|-------|-------|-------|-------|
> | *Instruction-tuning CL*      (teacher GPT-4o)              | 78.6   | 38.5 | 26.0  | 72.0   | 71.2  | 80.2 | 66.4  | **84.3**  |
> | *Instruction-tuning CL*      (teacher Qwen2-72b)              | 79.2   | **39.0** | 25.0  | **73.1**    | **72.8**  | **82.6** | **68.5**  | 83.2  |
> | *Instruction-tuning CL*      (teacher Mixtral8x7b)              | 77.0   | 36.4 | 24.2  | 72.0    | 71.2  | 81.0 | 66.0  | 80.4  |
> | *baseline*                                  | 73.0   | 28.0   | 10.0  | 58.6        | 58.0  | 62.9 | 66.5  | 48.2  |
>
> *Student Qwen2-7b
>
> ---
>
> | Method                                       | GSM8K  | MATH   | AIME24 | MMLU-Redux  | OBQA  | CSQA  | PIQA  | SIQA  |
> |---------------------------------------------|--------|--------|--------|-------------|-------|-------|-------|-------|
> | *Instruction-tuning CL*      (teacher GPT-4o)              | 76.9   | 38.0 | 22.8  | 70.2    | 71.4  | 81.0 | 66.1  | 83.6  |
> | *Instruction-tuning CL*      (teacher Qwen2-72b)              | 78.1   | 37.2 | 21.8  | 70.4    | 71.0  | 80.5 | 67.5  | 82.1  |
> | *Instruction-tuning CL*      (teacher Mixtral8x7b)              | **79.6**  | **39.8** | **25.0**  | **72.8**    | **73.1**  | **82.4** | **69.0** | **84.6**  |
> | *baseline*                                  | 72.8   | 28.2   | 10.0  | 58.3        | 59.2  | 63.5 | 67.2  | 48.6  |
>
> *Student Mistral-7b
>
> ---
> **Please see the table in the following comment for gemma-2-9b**
>
> 3. Thank you for this proposal. In order to test the impact of output language on model tuning, we have reproduced some tests in the following table.
>
> | Method                                       | GSM8K  | MATH   | AIME24 | MMLU-Redux  |
> |---------------------------------------------|--------|--------|--------|-------------|
> | *Instruction-tuning CL*                     | 78.6   | **38.5** | 26.0  | **72.0**    |
> | *Instruction-tuning CL+CoT len*              | 79.2 | 38.1   | 26.0  | 71.9        |
> | *Instruction-tuning  only CoT len*                 | 74.8   | 32.7   | 21.0  | 68.6        |
> | *baseline*                                  | 73.8   | 30.0   | 16.0  | 60.2        |
>
> *student Llama-3-8b and teacher GPT-4o
>
> As can be seen, the length of the outputs is not a very good indicator. Moreover, it does not change the final results much when combined with our heuristics. In order to make the analysis more robust, we will include these results in the final version.
>
> 4. Thank you for this observation. We have reported these results to Reviewer hrCx (in order not to be repetitive, we kindly ask you to see the reply to Reviewer hrCx). Please consider this analysis and if there is anything unclear, give us feedback, and we will be pleased to get back to you.
>
>
> 5. Thank you very much for bringing these unclear points to our attention. We promptly edited them and resolved the misleading parts (please see the correct sections in the updated PDF).
>
> In conclusion, thank you for your review and advice that helped us make the work more complete and solid. If there are any other points that are not clear, please let us know and we will be pleased to answer you.

---

> ### Author Response · Authors · 2025-05-03
> **Additional results**
>
> | Method                                       | GSM8K  | MATH   | AIME24 | MMLU-Redux  | OBQA  | CSQA  | PIQA  | SIQA  |
> |---------------------------------------------|--------|--------|--------|-------------|-------|-------|-------|-------|
> | *Instruction-tuning CL*      (teacher GPT-4o)              | 78.3   | 38.2 | 23.8  | 72.8    | 72.4  | **81.9** | 68.2  | **84.6**  |
> | *Instruction-tuning CL*      (teacher Qwen2-72b)              | 78.7   | **38.5** | 22.3  | **73.0**    | 71.0  | 80.5 | 67.5  | 82.1  |
> | *Instruction-tuning CL*      (teacher Mixtral8x7b)              | 79.0  | 37.8 | **24.2**  | 71.0    | 70.1  | 80.2 | 67.1 | 83.0  |
> | *baseline*                                  | 74.2   | 29.6   | 9.8  | 59.5        | 59.0  | 64.2 | 63.8  | 49.4  |
>
> *Student gemma-2-9b

---

### Decision · Action_Editor_zf4i · 2025-08-03

**Recommendation:** Accept as is

**Audience:**

Yes

**Audience Explanation:**

TMLR’s readership will value the practical for improving small and open models’ reasoning. The paper provides actionable guidance: curriculum-based ordering helps over standard instruction tuning, and in-family teacher–student alignment often yields stronger transfer.

**Claims And Evidence:**

Yes

**Claims Explanation:**

The paper’s central claims that (i) curriculum-guided instruction tuning (CL-IT) with LLM-generated chain-of-thought improves SLM reasoning, and (ii) using a teacher from the same model family tends to work better, are empirically supported. The authors report consistent gains of CL-IT over vanilla instruction tuning across multiple QA benchmarks and add follow-up results on math-oriented datasets. They also include ablations that compare their curriculum metrics against simpler ordering heuristics. Although the methodological novelty is modest and the scope leans toward multiple-choice QA, the evidence is adequate for the stated empirical claims.